# CoSE: Compositional Stroke Embeddings

**Emre Aksan**
ETH Zurich
eaksan@inf.ethz.ch

**Thomas Deselaers**\*
Apple Switzerland
deselaers@gmail.com

**Andrea Tagliasacchi**
Google Research
atagliasacchi@google.com

**Otmar Hilliges**
ETH Zurich
otmar.hilliges@inf.ethz.ch

## Abstract

We present a generative model for complex free-form structures such as stroke-based drawing tasks. While previous approaches rely on sequence-based models for drawings of basic objects or handwritten text, we propose a model that treats drawings as a collection of strokes that can be composed into complex structures such as diagrams (e.g., flow-charts). At the core of the approach lies a novel auto-encoder that projects variable-length strokes into a latent space of fixed dimension. This representation space allows a relational model, operating in latent space, to better capture the relationship between strokes and to predict subsequent strokes. We demonstrate qualitatively and quantitatively that our proposed approach is able to model the appearance of individual strokes, as well as the compositional structure of larger diagram drawings. Our approach is suitable for interactive use cases such as auto-completing diagrams. We make code and models publicly available at `https://eth-ait.github.io/cose`.

## 1 Introduction

Sketches and drawings have been at the heart of human civilization for millennia. While free-form sketching is a powerful and flexible tool for humans, it is a surprisingly hard task for machines, especially if interpreted in the generative sense. Consider Figure 1: *"when given only a sparse set of strokes (in black), what is the most likely continuation of a sketch or diagram (colored strokes are predicted)?"* The answer to this question is highly context sensitive and requires reasoning at the local (i.e., stroke) and global (i.e., the diagram or sketch) level.

Existing work has been focused on the recognition [1–3] and generation of handwritten text [4, 5] or the modelling of entire drawings [6–9] from the *Quick, Draw!* dataset [10]. However, the more recent `DiDi` dataset introduced by Gervais et al. [11], consisting of much more realistic and challenging *complex structures* such as diagrams and flow-charts, has been shown to be challenging for existing methods [12], due to the combinatorially many ways individual strokes can be combined into a complex drawing (see Fig. 7).

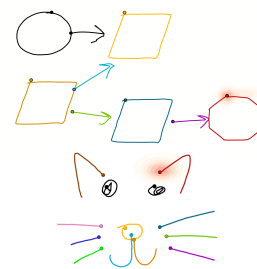

Figure 1: **Teaser** – We model complex drawings as a collections of strokes. Given only sparse strokes as input (black) the model predicts the most likely next strokes and their starting positions (heatmap), each color corresponds to one prediction step. This functional decomposition allows for generative modelling of varied and complex structures such as flow-charts (top), or freehand sketches (bottom). The drawings are model outputs.

In this paper we propose a novel *compositional* generative model, called CoSE, for complex stroke based data such as drawings, diagrams and sketches. While existing work considers the entire drawing as a single temporal sequence [3, 6, 7], our key insight is to factor *local* appearance of a stroke from the *global* structure of the drawing. To this end we treat each stroke as an *ordered* sequence of 2D positions $\mathbf{s}=\{(x_t, y_t)\}_{t=0}^{T}$, where $(x, y)$ represents the 2D location on screen. Importantly we treat the entire drawing $\mathbf{x}$ as an *unordered* collection of strokes $\mathbf{x}=\{\mathbf{s}_k\}_{k=1}^{K}$. Since the stroke ordering does not impact the semantic meaning of the diagram, this modelling decision has profound implications. In our approach the model does not need to understand the difference between the $(K-1)!$ potential orderings of the previous strokes to predict the $k$-th stroke, leading to a much more efficient utilization of modelling capacity. To achieve this we propose a generative model that first projects variable-length strokes into a latent space of fixed dimension via an encoder. A relational model then predicts the embeddings of future strokes, which are then rendered by the decoder.

The whole network is trained end-to-end and we experimentally show that the architecture can model complex diagrams and flow-charts from the `DiDi` dataset, free-form sketches from `QuickDraw` and handwritten text from the `IAM-OnDB` datasets [13]. We demonstrate the predictive capabilities via a proof-of-concept interactive demo (video in supplementary) in which the model suggests diagram completions based on initial user input. We show that our model outperforms existing models quantitatively and qualitatively and we analyze the learned latent space to provide insights into how predictions are formed.

## 2   Related Work

The interpretation of stroke data has been pursued before deep learning, often on small datasets of a few hundred samples targeting a particular application: Costagliola et al. [14] presented a parsing-based approach using a grammar of shapes and symbols where shapes and symbols are independently recognized and the results are combined using a non-deterministic grammar parser. Bresler et al. [15, 16] investigated flowchart and diagram recognition using a multi-stage approach including multiple independent segmentation and recognition steps.

For handwriting recognition, neural networks have been successfully used since Yaeger at al. [1] and LSTMs have been shown to be quite successful [4]. Recently, [17] have applied graph attention networks to 1,300 diagrams from [14–16] for text/non-text classification using a hand-engineered stroke feature vector. Yang et al. [8] apply graph convolutional networks for semantic segmentation at the stroke level to extensions of the `QuickDraw` data [18, 19]. For an in-depth treatment of drawing recognition, we refer the reader to the recent survey by Xu et. al. [9].

Particularly relevant to our work are approaches that apply generative models to stroke data. Ha et. al. [6] and Ribeiro et. al. [7] build LSTM/VAE-based and Transformer-based models respectively to generate samples from the `QuickDraw` dataset [20]. These approaches model the entire drawing as a single sequence of points. The different categories of drawings are modelled holistically without taking their internal structure into account. Graves proposed an auto-regressive handwriting generation model with LSTMs [21]; it explicitly models the sequence structure, hence making a full-sequence representation of the ink data a reasonable choice. In [5], an auto-regressive latent variable model is used to control the content and style aspects of handwriting, allowing for style transfer and synthesis applications.

Existing work either models the whole drawing (as an image) or as a complete sequence of points. In contrast, we model stroke-based drawings as order invariant 2D compositional structures and in consequence our model scales to more complex settings. Albeit in diverse and different domains, the following works are also relevant to ours in terms of explicitly considering the compositional nature of the problems.

Ellis et al. [22] use a program synthesis approach to analyze drawing images by recognizing primitives and combining them through program synthesis. This approach models the structure between components, but unlike our approach the model is applied to dense image-data rather than sparse strokes directly. One approach that applies neural networks to understand complex structures based on embeddings of basic building blocks is Lee et al. [23]. They learn an embedding space for mathematical equations and use a higher-level model to predict valid transformations of equations.

Wang et al. [24] follow an iterative approach to synthesize indoor scenes where the model picks an object from a database and decides where to place it. LayoutGAN [25] learns to generate realistic layouts from 2D wireframes and semantic labels for documents and abstract scenes.

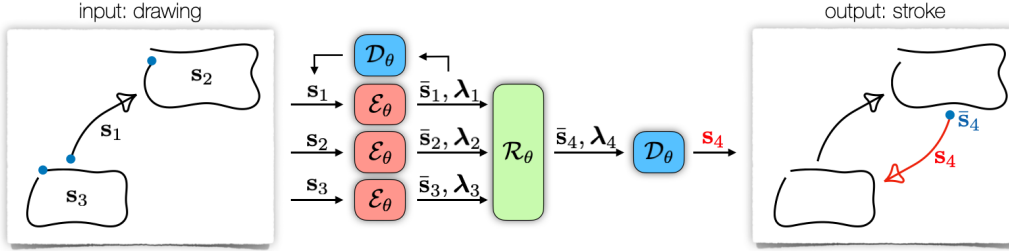

Figure 2: **Architecture overview** – (left) the input drawing as a collection of strokes $\{\mathbf{s}_k\}$; (middle) our embedding architecture, consisting of a shared encoder $\mathcal{E}_\theta$, a shared decoder $\mathcal{D}_\theta$, and a relational model $\mathcal{R}_\theta$; (right) the input drawing with the next stroke $\mathbf{s}_4$ and its starting position $\bar{\mathbf{s}}_4$ predicted by $\mathcal{R}_\theta$ and decoded by $\mathcal{D}_\theta$. Note that the relational model $\mathcal{R}_\theta$ is permutation-invariant.

## 3 Method

We are interested in modelling a drawing $\mathbf{x}$ as a collection of strokes $\{\mathbf{s}_k\}_{k=1}^K$, in the following abbreviated as $\{\mathbf{s}_k\}$, which requires capturing the semantics of a sketch and learning the relationships between its strokes. We propose a generative model, dubbed CoSE, that first projects variable-length strokes into a fixed-dimensional latent space, and then models their relationships in this latent space to predict future strokes. This approach is illustrated in Figure 2. More formally, given an initial set of strokes (*e.g.* $\{\mathbf{s}_1, \mathbf{s}_2, \mathbf{s}_3\}$), we wish to predict the next stroke (*e.g.* $\mathbf{s}_4$). We decompose the joint distribution of the sequence of strokes $\mathbf{x}$ as a product of conditional distributions over the set of existing strokes:

$$p(\mathbf{x}; \theta) = \prod_{k=1}^{K} p(\mathbf{s}_k, \bar{\mathbf{s}}_k | \mathbf{s}_{<k}, \bar{\mathbf{s}}_{<k}; \theta), \tag{1}$$

with $\bar{\mathbf{s}}_k$ referring to the starting position of the $k$-th stroke, and $<k$ denotes $\{1 \dots k-1\}$. Note that we assume a fixed but not chronological ordering of $K$. An encoder $\mathcal{E}_\theta$ first encodes each stroke $\mathbf{s}$ to its corresponding latent code $\boldsymbol{\lambda}$. A decoder $\mathcal{D}_\theta$ reconstructs the corresponding $\mathbf{s}$, given a code $\boldsymbol{\lambda}$ and the starting position $\bar{\mathbf{s}}$. A transformer-based relational model $\mathcal{R}_\theta$ processes the latent codes $\{\boldsymbol{\lambda}_{<k}\}$ and their corresponding starting positions $\{\bar{\mathbf{s}}_{<k}\}$ to generate the next stroke starting position $\bar{\mathbf{s}}_k$ and embedding $\boldsymbol{\lambda}_k$, from which $\mathcal{D}_\theta$ reconstructs the output stroke $\mathbf{s}_k$. Overall, our architecture factors into a *stroke embedding* model ($\mathcal{E}_\theta$ and $\mathcal{D}_\theta$) and a *relational model* ($\mathcal{R}_\theta$).

**Stroke embedding – Section 3.1.** We force the embedding model to capture *local* information such as the shape, size, or curvature by preventing it from accessing *any* global information such as the canvas position or existence of other strokes and their inter-dependencies. The auto-encoder generates an abstraction of the variable-length strokes $\mathbf{s}$ by encoding them into fixed-length embeddings $(\boldsymbol{\lambda}, \bar{\mathbf{s}}) = \mathcal{E}_\theta(\mathbf{s})$ and decoding them into strokes $\mathbf{s} = \mathcal{D}_\theta(\boldsymbol{\lambda}, \bar{\mathbf{s}})$.

**Relational model – Section 3.2.** Our relational model learns how to *compose* individual strokes to create a sketch by considering the relationship between latent codes. Given an input drawing encoded as $\mathbf{x} = \{(\boldsymbol{\lambda}_{<k}, \bar{\mathbf{s}}_{<k})\}$, we predict: i) a starting position for the next stroke $\bar{\mathbf{s}}_k$, and ii) its corresponding embedding $\boldsymbol{\lambda}_k$. Introducing the embeddings into Eq. 1, we obtain our *compositional stroke embedding* model that decouples *local* drawing information from *global* semantics:

$$p(\mathbf{x}; \theta) = \prod_{k=1}^{K} p(\boldsymbol{\lambda}_k, \bar{\mathbf{s}}_k | \boldsymbol{\lambda}_{<k}, \bar{\mathbf{s}}_{<k}; \theta) \tag{2}$$

We train by maximizing the log-likelihood of the network parameters $\theta$ on the training set.

### 3.1 Stroke embeddings

We represent variable-length strokes $\mathbf{s}$ with fixed-length embeddings $\boldsymbol{\lambda} \in \mathbb{R}^D$. The goal is to learn a representation space of the strokes such that it is informative both for *reconstruction* of the original strokes, and for *prediction* of future strokes. We now detail our auto-encoder architecture: the encoder $\mathcal{E}_\theta$ is based on transformers [26], while the decoder $\mathcal{D}_\theta$ extends ideas from neural modeling of differential geometry [27]. The parameters of both networks are trained via:

$$\arg\max_{\theta} \; \mathbb{E}_{t \sim [0,1]} \sum_{m=1}^{M} \pi_{t,m} \, \mathcal{N}(\mathbf{s}_k(t) \, | \, \mu_{t,m}, \sigma_{t,m}), \qquad \{\mu_{t,m}, \sigma_{t,m}, \pi_{t,m}\} = \mathcal{D}_\theta(t | \mathcal{E}_\theta(\mathbf{s})) \tag{3}$$

where we use mixture densities [21, 28] with $M$ Gaussians with mixture coefficients $\pi$, mean $\mu$ and variance $\sigma$; $t \in [0, 1]$ is the curve parameter. Note that we use log-likelihood rather than Chamfer

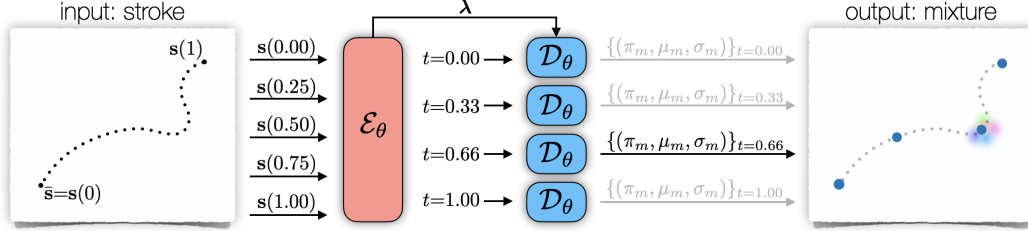

Figure 3: **Stroke embedding** – The input stroke **s** is passed to the encoder, which produces a latent code $\boldsymbol{\lambda}$. The decoder parameterizes a Gaussian mixture model for arbitrary positions $t{\in}[0, 1]$ from which we sample points on the stroke. We only visualize the mixture model associated with $t{=}.66$ (non-grayed out arrow).

Distance as in [27]. While we do interpret strokes as 2D curves, we observe that modelling of prediction uncertainty is commonly done in the ink modelling literature [5, 6, 21] and has been shown to result in better regression performance compared to minimizing an L2 metric [29] (cf. Sec. 10 in supplementary).

**CoSE Encoder –** $\mathcal{E}_\theta(\mathbf{s})$**.** We encode a stroke by viewing it as a sequence of 2D points, and generate the corresponding latent code with a transformer $\mathcal{T}_\theta^t$, where the superscript $t$ denotes use of positional encoding in the temporal dimension [26]. The encoder outputs $(\bar{\mathbf{s}}, \boldsymbol{\lambda}){=}\mathcal{E}_\theta(\mathbf{s})$, where $\bar{\mathbf{s}}$ is the starting position of a stroke and $\boldsymbol{\lambda}{=}\mathcal{T}_\theta^t(\mathbf{s} - \bar{\mathbf{s}})$. The use of positional encoding induces a point ordering and emphasizes the geometry, where most sequence models focus strongly on capturing the drawing dynamics. Furthermore, avoiding the modelling of explicit temporal dependencies between time-steps allows for inherent parallelism and is hence computationally advantageous over RNNs.

**CoSE Decoder –** $\mathcal{D}_\theta(t|\bar{\mathbf{s}}, \boldsymbol{\lambda})$**.** We consider a stroke as a curve in the differential geometry sense: a 1D manifold embedded in 2D space. As such, there exists a differentiable map $s : \mathbb{R} \to \mathbb{R}^2$ between $t{\in}[0, 1]$ and the 2D planar curve $s(t){=}(x_t, y_t)$. Groueix et al. [27] proposed to represent 2D (curves) and 3D (surfaces) geometry via MLPs that approximate $s(t)$. Recently it has been shown that representing curves via dense networks induces an implicit smoothness regularizer [30, 31] akin to the one that CNNs induce on images [32]. Since we do not want to reconstruct a single curve [31], we employ the latent code provided by $\mathcal{E}_\theta$ to condition our decoder jointly with the curve parameter $t$: $\mathcal{D}_\theta(t|\boldsymbol{\lambda}, \bar{\mathbf{s}}) = \bar{\mathbf{s}} + \mathrm{MLP}_\theta([t, \boldsymbol{\lambda}])$ [27] which parameterizes the Gaussian mixture from Eq. 3.

**Inference.** At inference time, a stroke is reconstructed by using $t$ values sampled at (consecutive) regular intervals as determined by an arbitrary sampling rate; see Figure 3. Note how compared to RNNs, we do not need to predict an "end of stroke" token, as the decoder output for $t{=}1$ corresponds to the end of a stroke. Therefore the length of the reconstructed sequence depends on how densely we sample the parameter $t$.

### 3.2 CoSE Relational model – $\mathcal{R}_\theta$

We propose a generative model that auto-regressively estimates a joint distribution over stroke embeddings and positions given a latent representation of the current drawing in the form $\mathbf{x}{=}\{(\boldsymbol{\lambda}_{<k}, \bar{\mathbf{s}}_{<k})\}$. We hypothesize that, in contrast to handwriting, local context and spatial layout are important factors that are not influenced by the drawing order of the user. We exploit the self-attention mechanism of the transformer [26] to learn the *relational dependencies* between strokes in the latent space. In contrast to the stroke embedding model (Sec. 3.1), we *do not* use positional encoding to prevent any temporal information to flow through the relational model.

**Prediction factorization.** In drawings, the starting position is an important degree of freedom. Hence, we split the prediction of the next stroke into two tasks: i) the prediction of the stroke's starting position $\bar{\mathbf{s}}_k$, and ii) the prediction of the stroke's embedding $\boldsymbol{\lambda}_k$. Given the (latent codes of) initial strokes, and their starting positions $\{(\boldsymbol{\lambda}_{<k}, \bar{\mathbf{s}}_{<k})\}$, this results in the factorization of the joint distribution over the strokes $\mathbf{s}_k$ and positions $\bar{\mathbf{s}}_k$ as a product of conditional distributions:

$$p(\boldsymbol{\lambda}_k, \bar{\mathbf{s}}_k|\boldsymbol{\lambda}_{<k}, \bar{\mathbf{s}}_{<k}; \theta) = \underbrace{p(\bar{\mathbf{s}}_k|\boldsymbol{\lambda}_{<k}, \bar{\mathbf{s}}_{<k}; \theta)}_{\text{starting position prediction}} \underbrace{p(\boldsymbol{\lambda}_k|\bar{\mathbf{s}}_k, \boldsymbol{\lambda}_{<k}, \bar{\mathbf{s}}_{<k}; \theta)}_{\text{latent code prediction}} \tag{4}$$

By conditioning on the starting position, the attention mechanism in $\mathcal{R}_\theta$ focuses on a *local* context, allowing our model to perform more effectively (see also Sec. 4.3). We use two separate transformers with the same network configuration yet slightly different inputs and outputs: i) the position prediction model takes the set $\{(\mathbf{s}_{<k}, \bar{\mathbf{s}}_{<k})\}$ as input and produces $\bar{\mathbf{s}}_k$; ii) the embedding prediction model takes the next starting position $\bar{\mathbf{s}}_k$ as additional input to predict $\boldsymbol{\lambda}_k$. Factorizing the prediction in this way

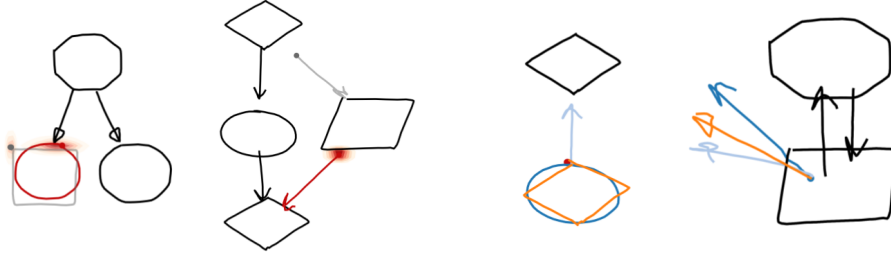

Figure 4: **Relational model** – A few snapshots from our live demo. (Left) Given a drawing, our model proposes *several* starting position for auto-completion (we draw the most likely strokes associated with the two most likely starting positions (red, gray)). (Right) Given a stating position, our model can predict *several* stroke alternatives; here we show the top 3 most likely predictions (orange, light blue, dark blue).

has two advantages: i) all strokes start at the origin, hence we can employ the translational-invariant embeddings from Sec. 3.1; ii) it enables interactive applications, where the user specifies a starting position and the model predicts an auto-completion; see video in the supplementary and Figure 4.

**Starting position prediction.** The prediction of the next starting positions is inherently multi-modal, since there may be multiple equally good predictions in terms of drawing continuation; see Figure 4 (left). We employ multi-modal predictions in the form of a 2-dimensional Gaussian Mixture. In the fully generative scenario, we sample a position $\bar{\mathbf{s}}_k$ from the predicted GMM, rather than expecting a user input or ground-truth data as at training time.

**Latent code prediction.** Given a starting position, multiple strokes can be used to complete a given drawing; see Figure 4 (right). We again use a Gaussian Mixture to capture this multi-modality. At inference time, we sample from $p(\boldsymbol{\lambda}_k|\bar{\mathbf{s}}_k, \boldsymbol{\lambda}_{<k}, \bar{\mathbf{s}}_{<k}; \theta)$ to generate $\boldsymbol{\lambda}_k$. Thanks to order-invariant relational model, CoSE can predict over long prediction horizons (see Fig. 5).

### 3.3 Training

Given a random pair of target $(\boldsymbol{\lambda}_k, \bar{\mathbf{s}}_k)$ and a subset of inputs $\{(\boldsymbol{\lambda}_{\neq k}, \bar{\mathbf{s}}_{\neq k})\}$, we make a prediction for the position and the embedding of the target stroke. This subset is obtained by selecting $H \in [1, K]$ strokes from the drawing. We either pick $H$ strokes i) in order, or ii) at random. This allows the model to be good in completing existing partial drawings but also be robust to arbitrary subsets of strokes. During training, the model has access to the ground-truth positions $\bar{\mathbf{s}}$ (like teacher forcing [33]). Note that while we train all three sub-modules (encoder, relational model, decoder) in parallel, we found that the performance is slightly better if gradients from the relational model (Eq. 2), are not back-propagated through the stroke embedding model. We apply augmentations in the form of random rotation and re-scaling of the entire drawing (see supplementary for details).

## 4 Experiments

We evaluate our model on the recently released `DiDi` dataset [11]. In contrast to existing drawing [20] or handwriting datasets [13], this task requires learning of the *compositional structure* of flowchart diagrams, consisting of several shapes. In this paper we focus on the predictive setting in which an existing (partial) drawing is extended by adding more shapes or by connecting already drawn ones. State-of-the-art techniques in ink modelling treat the entire drawing as a *single* sequence. Our experiments demonstrate that this approach does not scale to complex structures such as flowchart diagrams (cf. Fig. 7). We compare our method to the state-of-the-art [6] via the Chamfer Distance [34] between the ground-truth strokes and the model outputs (*i.e.* reconstructed or predicted strokes).

The task is inherently stochastic as the next stroke highly depends on where it is drawn. To account for the high variability in the predictions across different generative models, the ground-truth starting positions are passed the models in our quantitative analysis (note that the qualitative results rely only on the predicted starting positions). Moreover, similar to most predictive tasks, there is no single, correct prediction in the stroke prediction task (see Fig. 4). To account for this multi-modality of fully generative models, we employ a *stochastic* variant of the Chamfer Distance (CD):

$$\min_{\boldsymbol{\lambda}_k \sim p(\boldsymbol{\lambda}_k|\bar{\mathbf{s}}_k, \boldsymbol{\lambda}_{<k}, \bar{\mathbf{s}}_{<k}; \theta)} \left\{ \text{CD}\left(\mathcal{D}_\theta(t|\hat{\boldsymbol{\lambda}}_k), \mathbf{s}_k\right) \right\}. \tag{5}$$

We evaluate our models by sampling one $\boldsymbol{\lambda}_k$ from each mixture component of the relational model's prediction which are decoded into 10 strokes (see Fig. 9). This results in a broader exploration of the

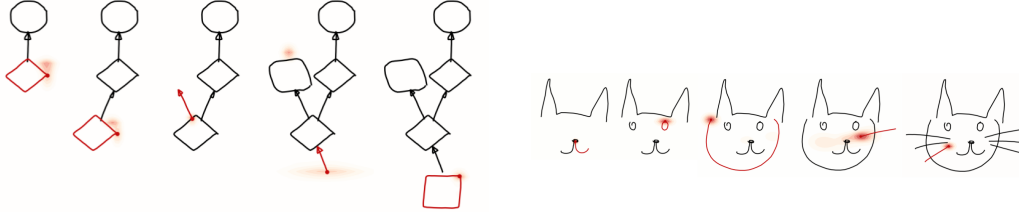

Figure 5: **Auto-regressive completion.** Performed by CoSE trained on `DiDi` and `QuickDraw` datasets.

predicted strokes than a strict Gaussian mixture sampling. Note that while our training objective is NLL (as is common in ink modelling), the Chamfer Distance allows for a fairer comparison since it allows to compare models trained on differently processed data (i.e., positions vs offsets).

## 4.1 Stroke prediction

We first evaluate the performance in the *stroke prediction* setting. Given a set of strokes and a target position, the task is to predict the next stroke. For each drawing, we start with a single stroke and incrementally add more strokes from the original drawing (in the order they were drawn) to the set of given strokes and predict the subsequent one. In this setting we evaluate our method in an ablation study, where we replace components of our model with standard RNN-based models: a sequence-to-sequence (seq2seq) architecture [35] for stroke-embeddings, and an auto-regressive RNN for the relational model (see supplementary for architecture details). Furthermore, following the setting in [6], we compare to the decoder-only setup from Sketch-RNN (itself conceptually similar to Graves et al. [21]). For the seq2seq-based embedding model we use bi-directional LSTMs [36] as the encoder, and a uni-directional LSTM as decoder. Informally we determined that a deterministic encoder with a non-autoregressive decoder outperformed other seq2seq architectures; see Sec. 4.2. The RNN-based relational model is an auto-regressive sequence model [21].

**Analysis.** The results are summarized in Table 1. While the stroke-wise reconstruction performance across all models differs only marginally, the predictive performance of our proposed model is substantially better. This indicates that a standard seq2seq model is able to learn an embedding space that is suitable for accurate reconstruction, this embedding space however *does not* lend itself to predictive modelling. The combination of our embedding model (CoSE-$\mathcal{E}_\theta/\mathcal{D}_\theta$) with our relational model (CoSE-$\mathcal{R}_\theta$) outperforms all other models in terms of predicting consecutive strokes, giving an indication that the learned embedding space is better suited for the predictive downstream tasks. The results also indicate that the contributions of both are necessary to attain the best performance. This can be seen by the increase in prediction performance of the seq2seq when augmented with our relational model (CoSE-$\mathcal{R}_\theta$). However, a significant gap remains to the full model (cf. row 2 and 5).

We also evaluate our full model with additional positional encoding in the relational model (CoSE + TR (Ordered)). The results support our hypothesis that an order-invariant model is beneficial for the task of modelling compositional structures. It is also observed in sequential modelling of the stroke embeddings by using an RNN (row 3). Similarly, our model outperforms Sketch-RNN which treats drawings as sequence. We show a comparison of flowchart completions by Sketch-RNN and CoSE in Fig. 7. Our model is more robust to make longer predictions.

## 4.2 Stroke embedding

Our analysis in Sec. 4.1 revealed that good reconstruction accuracy is not necessarily indicative of an embedding space that is useful for fully auto-regressive predictions. We now investigate the structure of our embedding space in qualitative and quantitative measures by analyzing the performance of clustering algorithms on the embedded data. Since there is only a limited number of shapes that

Table 1: **Stroke prediction** – We evaluate reconstruction (*i.e.* CD($\mathcal{D}_\theta(\mathcal{E}_\theta(\mathbf{s}))$, $\mathbf{s}$) and prediction (*i.e.* CD($\mathcal{R}_\theta(\lambda_{<k})$, $\mathbf{s}_k$) for a number of different models. Note that performing well on reconstruction does not necessarily correlate with good prediction performance.

| $\mathcal{E}_\theta/\mathcal{D}_\theta$ | $\mathcal{R}_\theta$ | Recon. CD↓ | Pred. CD↓ |
|---|---|---|---|
| seq2seq | RNN | 0.0144 | 0.0794 |
| seq2seq | CoSE-$\mathcal{R}_\theta$ | 0.0138 | 0.0540 |
| CoSE-$\mathcal{E}_\theta/\mathcal{D}_\theta$ | RNN | 0.0139 | 0.0713 |
| CoSE-$\mathcal{E}_\theta/\mathcal{D}_\theta$ | CoSE-$\mathcal{R}_\theta$ (Ord.) | 0.0143 | 0.0696 |
| CoSE-$\mathcal{E}_\theta/\mathcal{D}_\theta$ | CoSE-$\mathcal{R}_\theta$ | **0.0136** | **0.0442** |
| Sketch-RNN Decoder [6] | | N/A | 0.0679 |

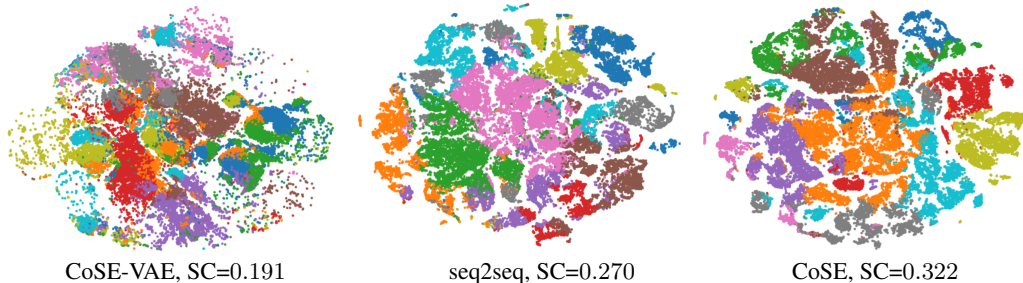

| CoSE-VAE, SC=0.191 | seq2seq, SC=0.270 | CoSE, SC=0.322 |

Figure 6: **tSNE Embedding** – Visualization of the latent spaces for different models (for quantitative analysis see Table 2). We employ $k$-means in latent space ($k{=}10$), and color by cluster ID. While a VAE regularized objective leads to an overall compact latent-space, clusters are not well separated, ours produces the most compact clusters (from left to right) which we show to be correlated with prediction quality.

occur in diagrams, the expectation is that a well shaped latent space should form clusters consisting of similar shapes, while maintaining sufficient variation.

**Silhouette Coefficient (SC).** This coefficient is a quantitative measure that assesses the quality of a clustering by jointly measuring tightness of exemplars within clusters vs. separation between clusters [37]. It does not require ground-truth cluster labels (*e.g.* whether a stroke is a box, arrow, arrow tip), and takes values between $[-1, 1]$ where a higher value is an indication of tighter and well separated clusters. The exact number of clusters is not known and we therefore compute the SC for the clustering result of $k$-means and spectral clustering [38] with varying numbers of clusters ($\{5, 10, 15, 20, 25\}$) with both Euclidean and cosine distance on the embeddings of all strokes in the test data. This leads to a total of 20 different clustering results. In Table 2, we report the average SC across these 20 clustering experiments for a number of different model configurations along with the Chamfer distance (CD) for stroke reconstruction and prediction. Note, the Pearson correlation between the SC and the prediction accuracy is 0.92 indicating a strong correlation between the two (see Sec. 8 in supplementary).

**Influence of the embedding dimensionality ($D$).** We performed experiments with different values of $D$ – the dimensionality of the latent codes. Table 2 shows that this parameter directly affects all components of the task: While a high-dimensional embedding space improves reconstructions accuracy, it is harder to predict valid embeddings in such a high-dimensional space and in consequence both the prediction performance and SC deteriorate. We observe a similar pattern with sequence-to-sequence architectures which benefit most from the increased embedding capacity by achieving the lowest reconstruction error (Recon. CD for seq2seq, D=32). However, it also leads to a significantly higher prediction error. Higher-dimensional embeddings result in less compact representation space, making the prediction task more challenging.

**Architectural variants.** In order to obtain a smoother latent space, we also introduce a KL-divergence regularizer [39] and follow the same annealing strategy as Ha et al. [6]. It is maybe surprising to see that a VAE regularizer (line CoSE-VAE) *hurts* reconstruction accuracy and interpretability of the embedding space. Note that the prediction task does not require interpolation or latent-space walks since latent codes represent entire strokes that can be combined in a discrete fashion. The results further indicate that our architecture yields a better behaved embedding space, while retaining a good reconstruction accuracy. This is indicated by i) the increase in reconstruction quality with larger $D$ yet prediction accuracy and SC deteriorate; ii) CoSE obtains much better prediction accuracy and SC at similar reconstruction accuracy; iii) smoothing the embedding space using a VAE for

Table 2: **Embedding space analysis** – (Top) Variants of our model with different embedding dimensionalities and a variant of our model with VAE. (Bottom) Results for a sequence-to-sequence stroke autoencoder (seq2seq) and its variational (VAE) and/or auto-regressive (AR) variants. All stroke embedding models use our Transformer relational model $\mathcal{R}_\theta$. $D$ indicates the dimensionality of the embedding space. CD and SC denote Chamfer Distance and Silhouette Coefficient, respectively.

| $\mathcal{E}_\theta/\mathcal{D}_\theta$ | D | Recon. CD $\downarrow$ | Pred. CD $\downarrow$ | SC $\uparrow$ |
|---|---|---|---|---|
| CoSE-$\mathcal{E}_\theta/\mathcal{D}_\theta$ (Tab. 1) | 8 | 0.0136 | **0.0442** | **0.361** |
| CoSE-$\mathcal{E}_\theta/\mathcal{D}_\theta$ | 16 | 0.0091 | 0.0481 | 0.335 |
| CoSE-$\mathcal{E}_\theta/\mathcal{D}_\theta$ | 32 | 0.0081 | 0.0511 | 0.314 |
| CoSE-$\mathcal{E}_\theta/\mathcal{D}_\theta$-VAE | 8 | 0.0198 | 0.0953 | 0.197 |
| seq2seq (Tab. 1) | 8 | 0.0138 | 0.0540 | 0.276 |
| seq2seq | 16 | 0.0076 | 0.0783 | 0.253 |
| seq2seq | 32 | **0.0047** | 0.0848 | 0.261 |
| seq2seq-VAE | 8 | 0.0161 | 0.0817 | 0.180 |
| seq2seq-AR | 8 | 0.0432 | 0.0855 | 0.249 |
| seq2seq-AR-VAE | 8 | 0.2763 | 0.1259 | 0.151 |

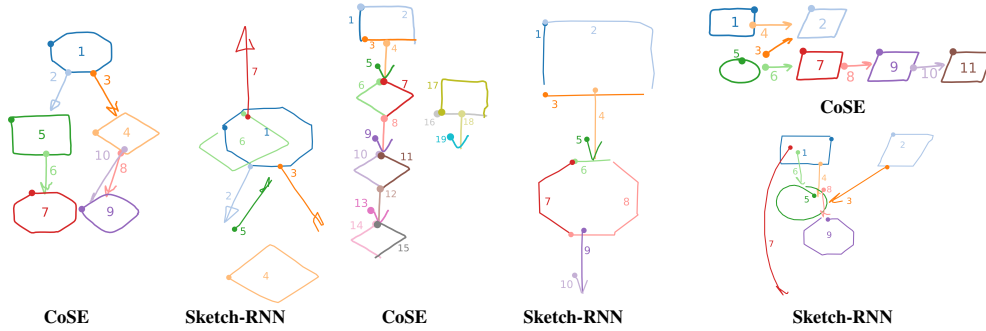

CoSE     Sketch-RNN     CoSE     Sketch-RNN     Sketch-RNN

Figure 7: **Comparison with Sketch-RNN** – For each pair of samples the first two strokes (in blue) are given as context, the remaining strokes (in color) are model outputs, numbers indicate prediction step. While Sketch-RNN produces meaningful completions for the first few predictions, its performance quickly decreases with increasing complexity. In contrast, CoSE is capable of predicting plausible continuations even over long prediction horizons.

regularization hurts reconstruction accuracy, prediction accuracy and SC; iv) autoregressive approach hurts reconstruction and prediction accuracy and SC - this is because autoregressive models tend to overfit to the ground-truth data (i.e., teacher forcing) and fail when forced to complete drawings based on their own predictions.

**Visualizations.** To further analyse the latent space properties, we provide a t-SNE visualization [40] of the embedding space with color coding for cluster IDs as determined by $k$-means with $k = 10$ in Figure 6. The plots indicate that the VAE objective encourages a latent space with overlapping clusters, whereas for CoSE, the clusters are better separated and more compact. An interesting observation is that the smooth and regularized VAE latent space does not translate into improved performance on either reconstruction or inference, which is inline with prior findings on the connection of latent space behavior and down-stream behavior [41]. Clearly, the embedding spaces learned using a CoSE model have different properties and are more suitable for predictive tasks that are conducted in the embedding space. This qualitative finding is inline with the quantitative results of the SC and correlating performance in the stroke prediction task (see also Sec. 9 in supplementary).

## 4.3 Ablations

**Conditioning on the start position.** The factorization in Eq. 4 allows our model to attend to a relatively local context. To show the importance of conditioning on the initial stroke positions, we train a model without this conditioning. Fig. 8 shows that conditioning on the start position helps to attend to the local neighborhood, which becomes increasingly important as the number of strokes gets larger. Moreover, the Chamfer Distance on the predictions nearly double from 0.0442 to 0.0790 in the absence of the starting positions.

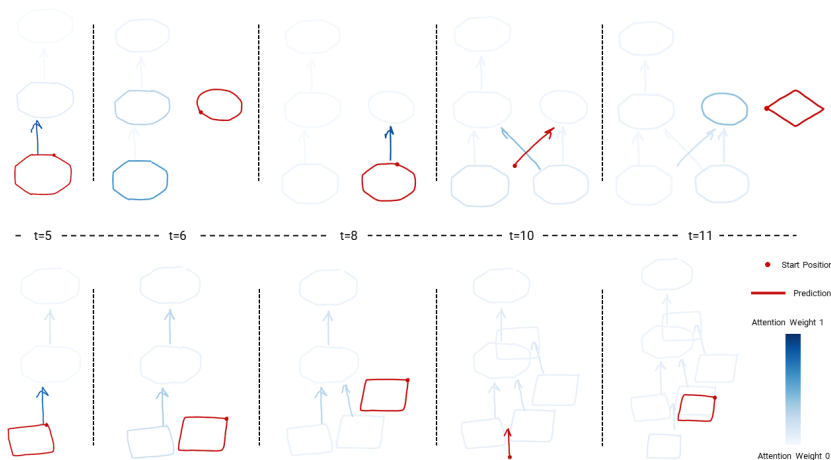

Figure 8: **Attention visualization over time** – (top) with and (bottom) without conditioning on the start position to make a prediction for the next stroke's (in red) embedding. Attention weights correspond to the average of all attention layers across the network.

**Number of GMM components.** Using a multi-modal distribution to model the stroke embedding predictions significantly improves our model's performance (cf. Fig. 9). It is an important hyper-parameter as it is the only source of stochasticity in our relational model $\mathcal{R}_\theta$. We observe that using 10 or more components is sufficient. Our results presented in the paper are achieved with 10 components.

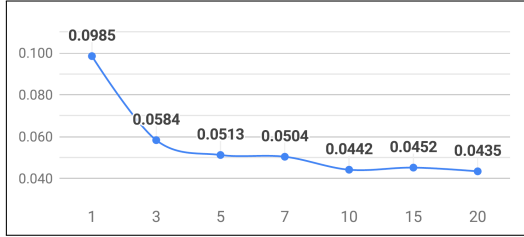

Figure 9: Pred. CD performance of our model $\mathcal{R}_\theta$ by using different number of components in the GMM for embedding predictions.

**Back-propagating $\mathcal{R}_\theta$ gradients.** Since we aim to decouple the local stroke information from the global drawing structure, we train the embedding model CoSE-$\mathcal{E}_\theta$/$\mathcal{D}_\theta$ via the reconstruction loss only, and do not back-propagate the relational model's gradients. We hypothesize that doing so would force the encoder to use some capacity to capture global semantics. When training our best model with all gradients flowing to the encoder CoSE-$\mathcal{E}_\theta$, increases the reconstruction error (Recon. CD) from 0.0136 to 0.0162 and the prediction error (Pred. CD) from 0.0442 to 0.0470.

## 4.4 Qualitative Results

The quantitative results from Table 1 indicate that our model performs better in the predictive modelling of complex diagrams compared to the baselines. Fig. 7 provides further indication that this is indeed the case. We show predictions of SketchRNN [6] which performs well on structures with very few strokes but struggles to predict more complex drawings. In contrast ours continues to produce meaningful predictions even for complex diagrams. This is further illustrated in Fig. 10 showing a number of qualitative results from a model trained on the `DiDi` (left), `IAM-OnDB` (center) and the `QuickDraw` (right) datasets, respectively. Note that all predictions are in the auto-regressive setting, where only the first stroke (in light blue) is given as input. All other strokes are model predictions (numbers indicate steps).

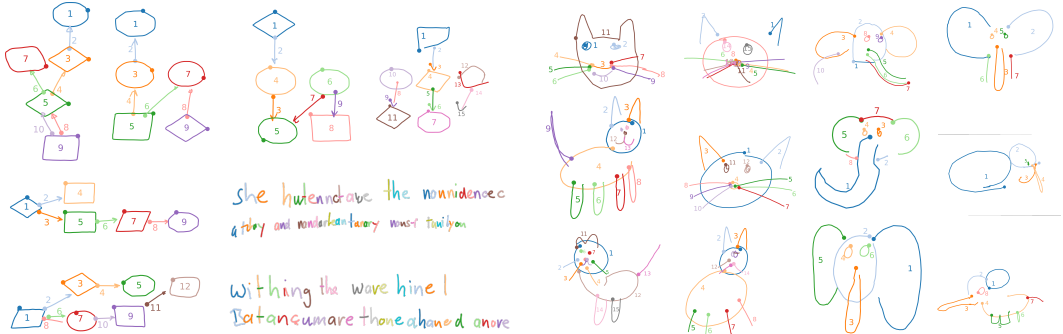

Figure 10: **Qualitative examples from CoSE** – Drawings were sampled from the model given the **first stroke**.

## 5  Conclusions

We have presented a *compositional generative model* for stroke data. In contrast to previous work, our model is able to model complex free-form structures such as those that occur in hand-drawn diagrams. A key insight is that by ignoring the ordering of individual strokes the complexity induced by the compositional nature of the data can be mitigated. This is achieved by decomposing our model into a novel auto-encoder that is able to create qualitatively better embedding spaces for predictive tasks and a relational model that operates on this embedding space. We demonstrate experimentally that our model outperforms previous approaches on complex drawings and provide evidence that a) good reconstruction accuracy is not necessarily indicative of good predictive performance b) the embedding space that our model learns strikes a good balance between these two and behaves qualitatively differently than the embedding spaces learned by the baselines models. We believe that in future work important concepts of our work can be directly applied to other tasks that have a compositional nature.

## Acknowledgments and Disclosure of Funding

The authors would like to thank Henry Rowley, Philippe Gervais, David Ha, and Andrii Maksai for their contributions to this work. This project has received funding from the European Research Council (ERC) under the European Union's Horizon 2020 research and innovation programme grant agreement No 717054 and from a Google Research Agreement.

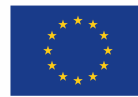 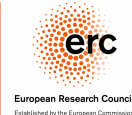

## Broader Impact

On the broader societal level, this work remains largely academic in nature, and does not pose foreseeable risks regarding defense, security, and other sensitive fields. One potential risk associated with all generative modelling work, is the danger of creating digital content that can be used for malicious purposes. In the context of stroke-based data the forgery of handwriting and signatures in particular is the most immediate concern. However, an attacker would require sufficient training data and extrapolation from such data would remain challenging. Furthermore, automating tasks that are commonly associated with creativity or craftsmanship holds some danger of rendering jobs or entire occupations redundant. However, we do believe that the positive impact of this work, which is to build technology that allows for better interaction between humans and computers by enabling for a more immediate and natural way to create structured drawings, will have a larger positive impact, such as improved creativity and better communication channels between humans.

## Footnotes

\*Work done while at Google. Unrelated to affiliation with Apple

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
