[Supplementary Material · cose_supplementary.pdf]

# CoSE: Compositional Stroke Embeddings
# (Supplementary Material)

**Emre Aksan**
ETH Zurich
eaksan@inf.ethz.ch

**Thomas Deselaers**
Apple Switzerland
deselaers@gmail.com

**Andrea Tagliasacchi**
Google Research
atagliasacchi@google.com

**Otmar Hilliges**
ETH Zurich
otmar.hilliges@inf.ethz.ch

In this supplementary material accompanying the paper *CoSE: Compositional Stroke Embeddings*, we provide the following:

- additional details about the model parameters and implementation. (Sec. 6, Sec. 7);
- further description about the correlation between the Silhouette Coefficient and predictive capabilities (Sec. 8);
- an additional visualization of the embedding space (Sec. 9);
- a discussion of MSE as reconstruction objective (Sec. 10);
- a discussion of the limitation of our model (Sec. 11);
- additional visualization of predictions from our model (Fig. 16).

In addition we also uploaded a video showing how a user interacts with our proposed model.

## 6 Data preparation and augmentation

The common representation for drawings employed in the related work is $\mathbf{x} = \{(x_t, y_t, p_t)\}_{t=1}^{T}$ – a sequence of temporarily ordered triplets $(x, y, p)$ of length $T$ corresponding to pen-coordinates on the device screen $(x, y)$ and pen-up events $p$ – i.e. $p{=}1$ when the pen is lifted off the screen, and $0$ otherwise [5, 21]. In [6], the pen-event is extended to triplets of pen-down, pen-up and end-of-sequence events.

We treat a segment of points up to a pen-up event (i.e., $p = 1$) as a stroke $\mathbf{s}$. Writing a letter or drawing a basic shape without lifting the pen results in a stroke. An ink sample can be considered as a sequence or set of strokes depending on the context. While it is of high importance to preserve the order in handwriting [3], we hypothesize that in free-form sketches, the exact ordering of strokes is not as important.

In this work, we consider strokes as an atomic unit instead of following a point-based representation as in previous work [5, 6, 21]. We define the ink sample as $\mathbf{x} = \{\mathbf{s}_i\}_{i=1}^{K}$ where $K$ is the number of strokes and the stroke $\mathbf{s}_i = \{(x_t, y_t)\}_{t=1}^{T_i}$ is a segment of $\mathbf{x}$. The stroke length $T_i$ is determined by the pen-up event.

We are using the data pre-processing as proposed by the `DiDi` dataset which consists of two steps: 1) normalizing the size of the diagram to unit height; 2) resampling the data along time with step size of 20 ms, meaning a stroke that took 1s to draw will be represented by 50 points. Each drawing is then stored as a variable length sequence of strokes. In the strokes, we only retain the $(x, y)$ coordinates.

During training, we apply a simple data augmentation pipeline of 3 steps for the drawing sample $\mathbf{x}$:

- Rotate the entire drawing with a random angle -90°and 90°.
- Scale the entire drawing by a random factor between 0.5 and 2.5
- Shear the entire drawing by a random factor between -0.3 and 0.3

Figure 11: **Transformer-based Stroke Encoders** Figures illustrate sequence- and set-based concepts for encoding the variable-length strokes **s** into **λ**. Each circle corresponds to a point in the stroke. Shading represents positional encoding to inject sequential information. Solid and dashed arrows denote access to information from past to the future and from the future to the past, respectively. (left) Our CoSE-$\mathcal{E}_\theta$ with using a *sequence* interpretation of strokes and (right) its counterpart using *sets*. Note that we stack a number of this layers in our model and the output of the top-most layer for the last input step is used as the stroke embedding **λ**.

where each of these steps is executed with a probability of 0.3.

# 7 Architecture Details

Our pipeline consists of a stroke embedding model with an encoder (CoSE-$\mathcal{E}_\theta$) and decoder ( CoSE-$\mathcal{D}_\theta$) and relational models to predict the stroke position and embeddings (CoSE-$\mathcal{R}_\theta$). In the paper we show results for an ablation experiment where we are replacing the individual components of our model with baselines, *i.e.*, we replace the stroke embedding with a seq2seq model, and we replace the relational model by a standard autoregressive RNN model. In the following we give details on the architectures used.

We aim to keep the number of trainable parameters similar in our experiments. All models have 2-3M parameters. We also run an hyper-parameter search for the reported hyper-parameters. Training of our full model takes around 20 hours on a GeForce RTX 2080 Ti graphics card.

We follow the notation of [26] to describe the hyper-parameters of Transformer-based building blocks.

## 7.1 CoSE-$\mathcal{E}_\theta$

Our encoder CoSE-$\mathcal{E}_\theta$ follows the Transformer encoder architecture [26]. We experiment with the following variants (see Fig. 11).

1. With positional encoding and look-ahead masks,
2. With positional encoding and without look-ahead masks in a bi-directional fashion,
3. Without positional encoding and without look-ahead masks, which corresponds to modelling the points in a stroke as set of points.

We empirically find that treating the stroke as a sequence rather than as a set yields significantly better performance. Similarly, restricting the attention operation from accessing the future points in the stroke is beneficial. This implies that the temporal information is informative for capturing local patterns in the stroke level. We further show that at the drawing level, the temporal information is not important.

For a given stroke sequence, we first feed it into a dense layer to get a $d_{model}$-dimensional representation for the transformer layers, which is followed by adding positional embeddings. We stack 6 layers with an internal representation size of $d_{model} = 64$ and feed-forward representation size of $d_{ff} = 128$. We use multi-head attention with 4 heads and omit dropout in the transformer layers.

To compute the stroke embedding **λ** we use the output of the top-most transformer layer at the last time-step. The stroke embedding **λ** is obtained by feeding this 64-dimensional vector into a single dense layer with $D$ output nodes without activation function.

## 7.2 CoSE-$\mathcal{D}_\theta$

Our decoder consists of 4 dense layers of size 512 with ReLU activation function. The output layer parameterizes a GMM with 20 components from which we then sample the $(x, y)$ positions for a given stroke embedding **λ** and $t$.

During training, we pair a stroke embedding **λ** with 4 random $t$ values and minimize the negative log-likelihood of the corresponding $s(t)$ targets. We map a stroke to the range $[0, 1]$ and then interpolate $s(t)$ for $t$ values that are not corresponding to a point.

### 7.3 CoSE-$\mathcal{R}_\theta$

We consider a drawing as a set of strokes and aim to model the relationship between strokes on a 2-dimensional canvas. We use the self-attention concept which explicitly relates inputs with each other. Our model follows the Transformer decoder architecture [26]. In order to prevent the model from having any sequential biases we do not apply positional encoding and look-ahead masks (similar to Fig. 11 right), enabling us to model strokes as a set.

We use separate models to make position and embedding predictions for the next stroke. We concatenate the given stroke embeddings and their corresponding start positions and feed into the position prediction model. The embedding prediction model additionally takes the start position of the next stroke. It is appended to the every given stroke. At inference time, we use the predicted start positions while they are the ground-truth positions during training. In summary, input size of the position prediction model is 10 consisting of 8D stroke embedding $\lambda$ and 2D position. For the embedding prediction model, it is 12 with an additional 2D start position of the next stroke. Similar to the CoSE-$\mathcal{E}_\theta$, the model representation of the top layer for the last input stroke is used to make a prediction. Both models predict a GMM with 10 components to model position and stroke embedding predictions.

We use the same configuration for both models. We stack 6 layers with an internal representation size of $d_{model} = 64$ and feed-forward representation size of $d_{ff} = 256$. We use multi-head attention with 4 heads and omit dropout in the transformer layers.

For a given drawing sample, we create 32 subsets of strokes randomly. 16 of them preserves the drawing order while the remaining 16 are shuffled (see Sec. 3.3).

Our model's sequential counterpart (i.e., CoSE-$\mathcal{R}_\theta$ (Ord.) in Tab. 1) uses positional encodings to access the temporal information. It is also trained by preserving the order of strokes in the input subsets.

### 7.4 Ablation models

We evaluate our hypothesis by replacing our model's components with fundamental architectures for the underlying task. In all experiments, we follow the same training and evaluation protocols as with our model.

#### 7.4.1 Seq2Seq Stroke Embedding Model

In our ablation study, we experiment with a seq2seq model to encode and decode variable-length strokes. We use LSTM cells of size 512 in the encoder and decoder. We find that processing the input stroke in both directions gives a better performance. For reconstruction, the decoder takes the stroke embedding at every step. In the seq2seq-AR counterpart, we feed the decoder with the prediction of the previous step as well.

Similar to our CoSE-$\mathcal{E}_\theta/\mathcal{D}_\theta$, the outputs are modeled with a GMM and the model is trained with negative log-likelihood objective.

#### 7.4.2 VAE Extension

We apply KL-divergence regularization to learn a smoother and potentially disentangled latent space. The encoder simply parameterizes a Normal distribution corresponding to the approximate posterior distribution. In addition to the reconstruction loss, we apply an KL-divergence loss between the approximate posterior and a standard Gaussian prior $\mathcal{N}(0, I)$. We follow the same annealing strategy as in [6].

#### 7.4.3 RNN Prediction Model

We use LSTM cells of size 512 for the position and embedding prediction models. We observe that training the RNN prediction models with both ordered and random subsets of stroke embeddings does not make a significant difference. In other works, the RNN models do not benefit from using set of strokes.

### 7.5 Sketch-RNN Baseline

We train the decoder-only Sketch-RNN model by following the instructions in [6] on the publicly-available codebase. We use an LSTM of size 1000 and use the default hyper-parameters. Similar to the quantitative evaluation of our models, we predict 10 samples from the Sketch-RNN model by

conditioning the model on the context strokes and ground-truth start position. We run the quantitative evaluation with various sampling temperatures in $[0.1, 0.3, 0.5, 0.7, 0.9]$ and report the best result.

## 7.6 Training

For the training of models with RNN components we find that an initial learning rate of $1e^{-3}$, annealed with an exponential decay strategy works the best. We use the Transformer learning rate scheduling proposed in [26] for the Transformer based models.

All models are trained with a batch size of 128. The stroke embedding models are trained by treating strokes independently. For the relational models, we create a new batch on-the-fly consisting of the random subsets of stroke embeddings.

# 8 Silhouette Coefficient

Fig. 12 plots the silhouette coefficient (SC) and prediction chamfer distance (CD) results reported in Table 2. As we note in the main paper, the Pearson correlation between the SC and the prediction metrics is 0.92 indicating a strong correlation between the two. We would like to note that we make this comparison among the models with our relation model CoSE-$\mathcal{R}_\theta$. Only the underlying embedding models vary.

Figure 12: Prediction Chamfer Distance (CD) (y-axis) and Silhouette Coefficient (SC) (x-axis) for the results presented in Table 2.

# 9 Embedding Predictions

In Fig. 13 we provide an additional analysis of the embedding spaces obtained from different models.

Each subplot shows a tSNE visualization of the embedding spaces, where blue points correspond to the embedding of strokes from the original data and yellow points correspond to embeddings predicted from our relational model $\mathcal{R}_\theta$. This clearly demonstrates that our model is able to predict much more "*natural*" embeddings in the embedding space from CoSE-$\mathcal{E}_\theta/\mathcal{D}_\theta$ than from the other two models. Our proposed model achieves a higher amount of overlapping with the real data. When we ignore our set assumption and model the strokes as a sequence, we start observing non-overlapping regions in the visualizations. Finally, our model fails to operate well in the latent space governed by the KL-divergence regularization. The underlying stroke embedding model is underfitting due to the strong KL-divergence regularizer, which further degrades the prediction performance. We also quantify this effect by calculating the Earth-Mover distance (EMD) between the two embedding distributions. Our model $CoSE$-$\mathcal{R}_\theta$ achieves an EMD of 155 while the sequential and the VAE counterparts result in EMDs of 1797 and 251, respectively. The EMD decreases as the GT and predicted distributions become more similar. We would like to note that our analysis on the embeddings also agrees with the prediction metric results reported in the main paper.

Figure 13: **tSNE Embedding** – Blue points correspond to embeddings computed from the original data, yellow points correspond to embeddings predicted by our relational model $\mathcal{R}_\theta$. (Left) Our model with KL-divergence regularization on the latent space (i.e., CoSE-$\mathcal{E}_\theta/\mathcal{D}_\theta$+VAE in Table 2), (middle) our model trained in a sequence-based fashion (i.e., CoSE-$\mathcal{R}_\theta$ (Ord.) in Tab. 1), (right) our model (i.e., CoSE-$\mathcal{R}_\theta$ in Tab. 1).

# 10 MSE Reconstruction Objective

We compare our probabilistic reconstruction objective (i.e., log-likelihood with a GMM) with a deterministic mean-squared error (MSE). The model configuration is the same in both experiments

Figure 14: **Reconstruction performance of deterministic and probabilistic objectives** (left) Ground-truth sample, reconstruction performance of the model trained with MSE (middle) and trained with log-likelihood objective and GMM output model (right).

with a difference in the reconstruction objective and the number of $t$ samples per stroke. We use $100$ $t$ samples for the model trained with MSE whereas it is only $4$ for the model with GMM predictions.

The MSE objective results in a competitive stroke reconstruction loss ($0.018$ compared to $0.014$ of the model with GMM log-likelihood). Fig. 14 shows qualitatively that the reconstructions are noisy although the overall shape information is preserved.

Figure 15: **Failure cases** – Given the first and second strokes, failed predictions of our model. (Left) A problem with connecting distant shapes via a long arrow. (Middle-Right) With increasing number of predictions, our model may predict overlapping arrows.

## 11 Limitations

In our work, we consider strokes as an atomic unit of drawings. However, there exist cases where users draw a very long and complex stroke such as in cursive handwriting. Our stroke embedding model fails to capture details of such long and complex strokes. We argue that it can be resolved by following a simple heuristic where too long strokes are split into shorter ones. Then the position prediction model is expected to predict the start position consecutive to the ending of the previous split.

In Fig. 15 we present common failure cases of our model. While it is straightforward to link the neighboring shapes, our model sometimes struggles to connect distant shapes via longer arrows.

We also observe that as the number of predicted strokes increases (i.e. $\sim 15$), it becomes more likely to predict arrow-like shapes by ignoring the existing content.

There are two main issues that may cause these failure cases. First, the number of shorter arrows connecting closer shapes is significantly higher in the training data. Hence, our model possibly fails to capture distant connection patterns. Second, our embedding space is not explicitly regularized to be smooth. Despite the fact that it works very well for our purpose, this may also cause our relational model to make predictions from regions with low density.

## 12 More visualization

In Fig. 16, we present more samples predicted by our model.

Figure 16: Given the **first** and **second** strokes, random predictions of our model.