[Reviews · NeurIPS 2020]

Review 1

Summary and Contributions: The paper describes a network that can generate or extend 2D drawings, without making the sequential structure assumptions of sketch2rnn. There is a sensible encoder/decoder architecture and relational network.

Strengths: The paper gets good results, with some reasonable ideas, on a signficant problem, i.e., modeling unstructured sketches. The qualitative results shown are compelling.

Weaknesses: A user interface is described, but no video is provided to give the reader a sense for how this works and how the interaction is.

Correctness: Yes, as far as I could tell.

Clarity: Yes.

Relation to Prior Work: There is a considerable amount of related work on generating unstructured layouts in other domains that arguably should be cited. See this paper for a survey: Learning Generative Models of 3D Structures Siddhartha Chaudhuri, Daniel Ritchie, Jiajun Wu, Kai Xu, and Hao Zhang Eurographics 2020 STAR as well as: PlanIT: Planning and Instantiating Indoor Scenes with Relation Graph and Spatial Prior Networks Kai Wang, Yu-an Lin, Ben Weissmann, Manolis Savva, Angel X. Chang, and Daniel Ritchie SIGGRAPH 2019 LayoutGAN: Synthesizing Graphic Layouts with Vector-Wireframe Adversarial Networks J. Li, J. Yang, A. Hertzmann, J. Zhang, T. Xu. IEEE Trans. Pattern Analysis and Machine Intelligence (PAMI). 2020.

Reproducibility: Yes

Additional Feedback: I remain positive that this is useful incremental advance in modeling collections of strokes. I can't comment on the novelty of the techniques to the world of machine learning as a whole, which was a concern for some other reviewers. I have lowered my score slightly for this reason, but I am in favor of acceptance nonetheless.


Review 2

Summary and Contributions: Post-rebuttal comments. The author's rebuttal addressed some of my concerns, and thus I updated my score accordingly. The paper introduces a generative model for stroke-based drawing using stroke embedding. The authors design an auto-encoder network as stroking embedding and a relational module to predict strokes based on input strokes. The key idea of this paper is to design an auto-encoder to embed a variable-length stroke with its starting position and a latent feature of fixed dimension. This stroke representation space also enables a relational model in latent feature space to model the relationship between strokes and to predict subsequent strokes. In this paper, the authors treat drawings as unordered stroke collections, and each stroke is regarded as a chronological ordered sequence of 2D positions.

Strengths: (1) The idea of factoring local appearance of a stroke from the global structure of the drawing is novel. (2) A stroke embedding module and a relational model are designed for capturing local information and relationship of strokes, respectively. Extensive experiments have been conducted to show the efficiency of this architecture. (3) This paper is well written.

Weaknesses: (1) My major concern about this work is the diversity of the predicted future strokes for input strokes, because there exists many combinations of the predicted strokes that are hard to generate. Although the paper adopts Guassian Mixture for the starting position and the latent code prediction (line 169-177), the random dynamic of strokes is surely still hard to model using the proposed method. (2) Due to the random dynamic of strokes, it is hard to define whether a specific prediction of the next stroke of a drawing is good or bad, especially for flowcharts. (3) The proposed method can only get sampled position of strokes, thus the continuity between strokes may be destroyed. For example, in the right part of Fig.5, the ears and face of the cat head are disconnected. (4) Since one of the major difference between the proposed model and Sketch-RNN is that this model predicts the next start point. The performance of the model in terms of the starting point and the next stroke should be compared respectively. (5) Other comments: -Lacking ablation study of relation model (with/without relation model) -Lacking comparisons of the performance with the starting position and the latent code prediction of different dimensional Guassian Mixture -Lacking training details such as the training time

Correctness: Yes

Clarity: The writing of this paper is good

Relation to Prior Work: Yes

Reproducibility: Yes

Additional Feedback: Due to the randomness of stroke sketching, it seems difficult to predict the strokes in the next step. A good experimental design that can clearly demonstrate the advantage of stoke prediction function in this paper, would be helpful to increase the impact of this paper.


Review 3

Summary and Contributions: The paper introduces an autoregressive generative model based on an autoencoder for penstrokes, considered as sequences of keypoints. The paper particularly focuses on disentangling local aspects of individual strokes from the global, relational way in which strokes are composed to form a whole sketch.

Strengths: The paper makes strong contributions in representation learning for the strokes, and succeeds at reconstructing and predicting interesting diagrams and drawings. Despite an autoencoder setup, the paper focuses its experiments and evaluation on predicting the completion of a partial drawing.

Weaknesses: The paper could have been just a little more impressive by using some sort of neural or stochastic renderer to map from the stroke sequences to images and vice-versa. I would like to see comparisons to seq2seq models with equal embedding dimensionality to the COSE model, though this only appears in the paper for D=8.

Correctness: The claims and methods appear to be accurate, as far as I can tell. The empirical methodology is NeurIPS-quality.

Clarity: The paper is not only clear, but manages to nicely pass from the intuitive description of its target problem at the beginning to the technical methodology. The figure captions could have been somewhat clearer.

Relation to Prior Work: The paper neatly situates itself with respect to both image-based models, generative ink models, and program synthesis methods.

Reproducibility: Yes

Additional Feedback: The authors have addressed my worry about what happens if gradients propagate through the relational model.


Review 4

Summary and Contributions: This work presents a generative model for compositional sketches. The proposed model auto-regressively models strokes with a transformer to learn relations between the strokes. The proposed model is evaluated on the DiDi and Quick,Draw! datasets.

Strengths: The main strengths are, + the paper is well written and easy to understand. + the novelty with respect to prior work esp. Sketch-RNN Decoder [6] is clearly explained. + the qualitative examples in Fig. 7 clearly shows the benefit wrt to Sketch-RNN Decoder [6]. + the paper includes enough ablations to demonstrate the effectiveness of its various components.

Weaknesses: The rebuttal and discussion clarified my concerns about [1,2] (although I would highly encourage that these works be citied for a more complete related works section). However, I remain unconvinced by the novelty of the approach -- the fact that transformer based models work better compared to simple VAE based models is not surprising to the general NeurIPS audience. However, I do agree that from the point of view of stroke based generative models the work is novel and makes a good contribution to this specific field. ------------------------------------------------------------------------------- The main weakness are, - Limited novelty -- very similar models have been proposed in prior work [1]. Novelty wrt to [1] is not clear -- both methods use a transformer based architecture to model long-range dependencies in strokes. The advantage of an autoregressive structure along with transformers is not clear as transformers contain self-attention layers to capture long range dependencies. Even though the work [1] appears at CVPR 2020 and does not influence the rating of the proposed method, the evaluation metrics like mAP% [1] to evaluate the quality of the latent spaces should have been considered. - Limited baselines. The work should compare with recent works [1,2] and not be limited to the older SketchRNN baseline. An ideal dataset for comparison would be QuickDraw50M or Sketchy. Only qualitative results are provided with QuickDraw and a more detailed evaluation with the prior state of the art would be beneficial and highlight the advantages of the proposed method across datasets. [1] Sketchformer: Transformer-based Representation for Sketched Structure, CVPR 2020. [2] Synthesizing human-like sketches from natural images using a conditional convolutional decoder, WACV 2020.

Correctness: Yes.

Clarity: Yes,

Relation to Prior Work: No.

Reproducibility: Yes

Additional Feedback:

[Author Response · NeurIPS 2020]

We thank reviewers for their insightful comments and are happy that they find the problem significant (**R1**) and
challenging (**R2**), the proposed decomposition novel (**R2**) and the results compelling (**R1**). In the following, we address
concerns and provide new experimental evidence. Fig. 1-8 and Tab. 1-2 are in the main paper and Fig. 9-15 can be
found in the appendix. We kindly ask the reviewers and the AC zoom into the figures.

**R1: Suggested papers and Video**. Thank you for pointers to additional papers; we will
include a discussion. Note that a video *is* available in the *supplementary materials*.

**R2: Stochasticity of task**. Our task shares similarities to NLP problems such as text
auto-completion in gmail. In text prediction, localization hints are provided by
positional encoding, and the "starting position" is the last token; the attention model
in transformers allows the model to determine the relevant *local* context to predict
the next token. In drawings, on the other hand, the starting position is *not* fixed and

Figure 16: Average attention visualization over time with (top) and without (bottom) conditioning on the start position. Please enlarge Fig.

an important degree of freedom. Hence the attention model in CoSE-$\mathcal{R}_\theta$ allows the
prediction to focus on a *local* context by conditioning on the starting position. This allows our model to perform
effectively. To show the importance of the initial stroke positions, we trained a model without conditioning on them
and see the CD nearly double from $0.0442$ (Tab. 2) to $0.0790$ (new). Fig. 16 also shows that conditioning on the start
position helps to attend to the nearby strokes, which is increasingly important as the number of strokes gets larger.

**R2: Relational model ablation**. Note that predicting starting positions alone is not enough. A
crucial component in capturing pairwise dependencies is the proposed relational model CoSE-
$\mathcal{R}_\theta$. Performance degrades substantially if we replace CoSE-$\mathcal{R}_\theta$ with an LSTM, receiving
stroke embeddings in drawing order (Tab. 4). Sketch-RNN models the data as a sequence
of points in contrast to our compositional approach.

Table 4: Ablation on $\mathcal{R}_\theta$

| $\mathcal{E}_\theta/\mathcal{D}_\theta$ | $\mathcal{R}_\theta$ | CD↓ |
|---|---|---|
| CoSE-$\mathcal{E}_\theta/\mathcal{D}_\theta$ | CoSE-$\mathcal{R}_\theta$ | **0.0442** |
| CoSE-$\mathcal{E}_\theta/\mathcal{D}_\theta$ | RNN | 0.0713 |
| | Sketch-RNN | 0.0679 |

**R2: Diversity of the predictions**. Given an initial position, the GMM contains a diverse set
of predictions (Fig. 4). In Fig. 17, we ablate wrt the number of components as requested.
The ability of our model to generate similar diversity to the test set is also visible in Sec. 9:
mode collapse would incur a visible difference in the distribution of ground-truth (blue) vs.
predicted (yellow) embeddings (cf, Fig.11-left). We quantify this effect by calculating the
Earth-Mover distance (EMD) between the two embedding distributions. Fig.11, left-to-right:

Figure 17: Pred. CD vs. # GMM.

EMDs of $1797$, $251$ and $155$ (ours). The EMD decreases as the GT and predicted distributions become more similar.

**R2: Stroke discontinuity**. Note that this is emergent behavior from the dataset which contains many such examples.

**R2: Experimental design**. The results summarized in Fig. 16 & Tab. 4 show that modeling of pairwise dependencies and
predicting the next embedding are crucial. Our experiments assess different models under that assumption and we focus
on the task of predicting the next stroke giving a partial drawing. To control high variability in the predictions across
different generative models, we feed ground-truth starting positions in our quantitative analysis (note that the qualitative
results rely only on the *predicted* starting positions). We furthermore use a stochastic metric (Eq. 5) to ensure fairness.
Moreover, our final metric, the chamfer distance (CD) of the strokes, allows us to compare models trained with different
objectives (e.g., next point prediction as in SketchRNN) and different representations (e.g., velocity).

**R3: Gradients**. We aim to decouple the local stroke from the global drawing structure. We train via the *reconstruction*
loss only, and do not back-propagate the relational model's gradients. Doing so would force the encoder to use some
capacity to capture global semantics. Training our best model with all gradients flowing to the encoder, the error (Recon.
CD) increases from $0.0136$ to $0.0162$ and the prediction error (Pred. CD) from $0.0442$ to $0.0470$.

**R3: Embedding size**. We compare CoSE-$\mathcal{E}_\theta/\mathcal{D}_\theta$ and the baseline seq2seq with
varying embedding size; see Tab. 5. We use CoSE-$\mathcal{R}_\theta$ to evaluate the predictive
power of the corresponding embeddings. For both models, the reconstruction
performance improves with increasing embedding size. However, it also results in
a less compact representation space, making the prediction task *more challenging*.

Table 5: Ablation on embedding size $D$

| $\mathcal{E}_\theta/\mathcal{D}_\theta$ | D | Recon. CD↓ | Pred. CD↓ | SC↑ |
|---|---|---|---|---|
| CoSE-$\mathcal{E}_\theta/\mathcal{D}_\theta$ | 8 | 0.0136 | **0.0442** | **0.361** |
| CoSE-$\mathcal{E}_\theta/\mathcal{D}_\theta$ | 16 | 0.0091 | 0.0481 | 0.335 |
| CoSE-$\mathcal{E}_\theta/\mathcal{D}_\theta$ | 32 | 0.0081 | 0.0511 | 0.314 |
| seq2seq | 8 | 0.0138 | 0.0540 | 0.276 |
| seq2seq | 16 | 0.0076 | 0.0783 | 0.253 |
| seq2seq | 32 | **0.0047** | 0.0848 | 0.261 |

**R4: Novelty**. We respectfully disagree with **R4** on the limited novelty. We don't
simply replace RNNs with transformers but propose a novel task decomposition that we show to be important and
propose a novel architecture to capture stroke dependencies in an unordered fashion. Further, we quote from the
*official* reviewing guidelines that ``excuse authors for not knowing all non-refereed work (e.g, ArXiv)''. Both
references were recently published (2/3 months) on ArXiv at submission time (see below for differences).

**R4: Baselines**. Sketchformer learns sketch representations for image re-
trieval (SBIR) using full supervision whereas our task is fully unsupervised.
The suggested $mAP\%$ metric requires *labels* for evaluation. We emphasize that
our goal is to learn the compositions of strokes into drawings, rather than the
entire sketch, to allow for scalability wrt to sketch complexity.

Our approach can generalize to different domains, we provide qualitative results
on QuickDraw sketch (Fig. 5) and IamOnDB handwriting datasets (Fig. 18)

Figure 18: Given the **first** and second strokes, unconditional handwriting samples generated by our model.

[Meta-Review · NeurIPS 2020]

The paper initially received a mixed rating, but all the reviewers rate the paper above the bar after intensive discussion. While the reviewers agree that the paper may not present a groundbreaking algorithm, the design choices are all sensible, and the empirical results are convincing. After consolidating the reviews and rebuttal, the AC agrees with the assessment and would like to recommend the acceptance of the paper.